 SciPost Phys. Lect. Notes 21 (2021)

# Useful relations among the generators in the defining and adjoint representations of SU(N)

**Howard E. Haber**

Santa Cruz Institute for Particle Physics, University of California, Santa Cruz, CA 95064, USA

## Abstract

There are numerous relations among the generators in the defining and adjoint representations of $SU(N)$. These include Casimir operators, formulae for traces of products of generators, etc. Due to the existence of the completely symmetric tensor $d_{abc}$ that arises in the study of the $SU(N)$ Lie algebra, one can also consider relations that involve the adjoint representation matrix, $(D^a)_{bc} = d_{abc}$. In this review, we summarize many useful relations satisfied by the defining and adjoint representation matrices of $SU(N)$. A few relations special to the case of $N = 3$ are highlighted.

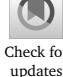

## 1 Introduction

The $SU(N)$ Lie group and its Lie algebra are ubiquitous in theoretical physics. Numerous relations among the generators in the defining and adjoint representations of $SU(N)$ are often

useful in a variety of physics applications. Many of these relations are well known and others are more obscure. There are multiple sources for the various identities that will be reviewed in these notes, but there is no single reference that I am aware of that contains all of them. For my own benefit, as well as for the benefit of others, I have collected many of the relevant identities and assembled them in this short review. In a few instances, I found typographical errors in some of the original sources that I was able to correct.

Recently, two authors contacted me concerning a first draft of these notes, which they apparently found to be quite useful [1,2]. They urged me to make these notes more widely available, and I am pleased to accommodate their request. Although I have not taken the opportunity for providing a more comprehensive list of references, I have included references to the primary sources that were used in obtaining all the formulae of this review.

## 2 The defining representation of the SU($N$) Lie algebra

In these notes, we provide some useful relations involving the generators of the SU($N$) Lie algebra, henceforth denoted by $\mathfrak{su}(N)$. We employ the physicist's convention, where the $N^2-1$ generators in the defining representation of $\mathfrak{su}(N)$, denoted by $T^a$, serve as a basis for the set of traceless hermitian $N \times N$ matrices. The generators satisfy the commutation relations,

$$[T^a, T^b] = if_{abc}T^c, \qquad \text{where } a, b, c = 1, 2, \ldots, N^2-1. \tag{1}$$

In particular

$$\operatorname{Tr} T^a = 0. \tag{2}$$

We employ the following normalization convention for the generators in the defining representation of $\mathfrak{su}(N)$,

$$\operatorname{Tr}(T^a T^b) = \tfrac{1}{2}\delta_{ab}. \tag{3}$$

In this convention, the $f^{abc}$ are totally antisymmetric with respect to the interchange of any pair of its indices.

Consider a $d$-dimensional irreducible representation, $R^a$ of the generators of $\mathfrak{su}(N)$. The quadratic Casimir operator, $C_2 \equiv R^a R^a$, commutes with all the $\mathfrak{su}(N)$ generators.[1] Hence in light of Schur's lemma, $C_2$ is proportional to the $d \times d$ identity matrix. In particular, the quadratic Casimir operator in the defining representation of $\mathfrak{su}(N)$ is given by

$$T^a T^a = C_F \mathbb{1}, \tag{4}$$

where $\mathbb{1}$ is the $N \times N$ identity matrix. To evaluate $C_F$, we take the trace of eq. (4) and make use of $\operatorname{Tr} \mathbb{1} = N$. Summing over $a$, we note that $\delta_{aa} = N^2-1$. Using the normalization of the generators specified in eq. (3), it follows that $\tfrac{1}{2}(N^2-1) = NC_F$. Hence,[2]

$$C_F = \frac{N^2-1}{2N}. \tag{5}$$

Next we quote an important identity involving the $\mathfrak{su}(N)$ generators in the defining representation,

$$T^a_{ij} T^a_{k\ell} = \frac{1}{2}\left(\delta_{i\ell}\delta_{jk} - \frac{1}{N}\delta_{ij}\delta_{k\ell}\right), \tag{6}$$

---

[1]It is straightforward to show that $C_2$ commutes with all the generators of $\mathfrak{su}(N)$. In particular, using the commutation relations, $[R^a, R^b] = if_{abc}R^c$,

$$[R^a R^a, R^b] = R^a[R^a, R^b] + [R^a, R^b]R^a = if^{abc}(R^a R^c + R^c R^a) = 0,$$

due to the antisymmetry of $f^{abc}$ under the interchange of any pair of indices.

[2]In the older literature, the defining representation is (inaccurately) called the fundamental representation. It is for this reason that the Casimir operator in the defining representation is often denoted by $C_F$.

where the indices $i$, $j$, $k$ and $\ell$ take on values from $1, 2, \ldots, N$. To derive eq. (6), we first note that any $N \times N$ complex matrix $M$ can be written as a complex linear combination of the $N \times N$ identity matrix and the $T^a$,

$$M = M_0 \mathbb{1} + M_a T^a . \tag{7}$$

This can be regarded as a completeness relation on the vector space of complex $N \times N$ matrices. One can project out the coefficient $M_0$ by taking the trace of eq. (7). Likewise, one can project out the coefficients $M_a$ by multiplying eq. (7) by $T^b$ and then taking the trace of the resulting equation. Using eqs. (2) and (3), it follows that

$$M_0 = \frac{1}{N} \operatorname{Tr} M , \qquad M_a = 2 \operatorname{Tr}(M T^a) . \tag{8}$$

Inserting these results back into eq. (7) yields

$$M = \frac{1}{N} (\operatorname{Tr} M) \mathbb{1} + 2 \operatorname{Tr}(M T^a) T^a . \tag{9}$$

The matrix elements of eq. (9) are therefore

$$M_{ij} = \frac{1}{N} M_{kk} \delta_{ij} + 2 M_{\ell k} T^a_{k\ell} T^a_{ij} , \tag{10}$$

where the sum over repeated indices is implicit. We can rewrite eq. (10) in a more useful form,

$$\delta_{i\ell} \delta_{jk} M_{\ell k} = \left( \frac{1}{N} \delta_{ij} \delta_{k\ell} + 2 T^a_{ij} T^a_{k\ell} \right) M_{\ell k} . \tag{11}$$

It follows that

$$\left[ T^a_{ij} T^a_{k\ell} - \frac{1}{2} \left( \delta_{i\ell} \delta_{jk} - \frac{1}{N} \delta_{ij} \delta_{k\ell} \right) \right] M_{\ell k} = 0 . \tag{12}$$

This equation must be true for any arbitrary $N \times N$ complex matrix $M$. It follows that the coefficient of $M_{\ell k}$ in eq. (12) must vanish. This yields the identity states in eq. (6). The proof is complete.

Many important identities can be obtained from eq. (6). For example, multiplying eq. (6) by $T^b_{jk}$ and summing over $j$ and $k$ yields

$$T^a T^b T^a = -\frac{1}{2N} T^a , \tag{13}$$

after employing eq. (2). If we now multiply eq. (13) by $T^c$ and take the trace of both sides of the resulting equation, then the end result is

$$\operatorname{Tr}(T^a T^b T^a T^c) = -\frac{1}{4N} \delta_{bc} , \tag{14}$$

after using eq. (3). A more general expression for the trace of four generators (of which eq. (14) is a special case) is given in Appendix A.

## 3  Introducing the symmetric third rank tensor $d_{abc}$

In $\mathfrak{su}(N)$, one can also define a totally symmetric third rank tensor called $d^{abc}$ via the relation,

$$T^a T^b = \frac{1}{2} \left[ \frac{1}{N} \delta_{ab} \mathbb{1} + (d_{abc} + i f_{abc}) T^c \right] , \tag{15}$$

where $\mathbb{1}$ is the $N \times N$ identity matrix. Combining eqs. (1) and (15) yields the following anti-commutation relation,

$$\{T^a, T^b\} \equiv T^a T^b + T^b T^a = \frac{1}{N}\delta_{ab}\mathbb{1} + d_{abc}T^c. \tag{16}$$

Using eqs. (3) and (16), one obtains an explicit expression,

$$d_{abc} = 2\operatorname{Tr}\left[\{T^a, T^b\}T^c\right], \tag{17}$$

which can be taken as the definition of the $d_{abc}$. It then follows that $d_{aac} = 0$ (where a sum over the repeated index $a$ is implicit). Indeed, since $d_{abc}$ is a totally symmetric tensor, it follows that $d_{aca} = d_{caa} = 0$.

The case of $\mathfrak{su}(2)$ provides the simplest example. In this case, we identify $T^a = \frac{1}{2}\sigma^a$, where the $\sigma^a$ (for $a = 1, 2, 3$) are the well-known Pauli matrices, and $f_{abc} = \epsilon_{abc}$ are the components of the Levi-Civita tensor. It is a simple matter to check that in the case of $\mathfrak{su}(2)$, we have $d_{abc} = 0$. In contrast, the $d_{abc}$ are generally non-zero for $N \geq 3$.

Consider the trace identity obtained by multiplying eq. (15) by $T^c$ and taking the trace. In light of eqs. (2) and (3),

$$\operatorname{Tr}(T^a T^b T^c) = \tfrac{1}{4}\left(d_{abc} + if_{abc}\right). \tag{18}$$

It then follows that

$$f_{abd}\operatorname{Tr}(T^a T^b T^c) = \tfrac{1}{4}if_{abc}f_{abd}, \tag{19}$$

$$d_{abd}\operatorname{Tr}(T^a T^b T^c) = \tfrac{1}{4}d_{abc}d_{abd}. \tag{20}$$

In obtaining eqs. (19) and (20), we used the fact that $d_{abc}$ is symmetric and $f_{abc}$ is antisymmetric under the interchange of any pair of indices, which implies that

$$f_{abc}d_{abd} = 0. \tag{21}$$

To evaluate the products $f_{abc}f_{abd}$ and $d_{abc}d_{abd}$, we proceed as follows. Using eqs. (1) and (16),

$$f_{abd}\operatorname{Tr}(T^a T^b T^c) = -i\operatorname{Tr}\left([T^b, T^d]T^b T^c\right) = -i\operatorname{Tr}(T^b T^d T^b T^c) + i\operatorname{Tr}(T^d T^b T^b T^c), \tag{22}$$

$$d_{abd}\operatorname{Tr}(T^a T^b T^c) = \operatorname{Tr}\left[\left(\{T^b, T^d\} - \frac{1}{N}\delta_{bd}\mathbb{1}\right)T^b T^c\right]$$

$$= \operatorname{Tr}(T^b T^d T^b T^c) + \operatorname{Tr}(T^d T^b T^b T^c) - \frac{1}{N}\operatorname{Tr}(T^d T^c). \tag{23}$$

The traces are easily evaluated using eqs. (3)–(5) and (14), and we end up with

$$f_{abd}\operatorname{Tr}(T^a T^b T^c) = \tfrac{1}{4}iN\delta_{cd}, \tag{24}$$

$$d_{abd}\operatorname{Tr}(T^a T^b T^c) = \left(\frac{N^2 - 4}{4N}\right)\delta_{cd}. \tag{25}$$

Comparing eqs. (24) and (25) with eqs. (19) and (20), we conclude that,[3]

$$f_{abc}f_{abd} = N\delta_{cd}, \tag{26}$$

$$d_{abc}d_{abd} = \left(\frac{N^2 - 4}{N}\right)\delta_{cd}. \tag{27}$$

---

[3]Note that eqs. (21), (26) and (27) are equivalent to eqs. (45) and (46), respectively.

Consider a $d$-dimensional irreducible representation, $R^a$ of the generators of $\mathfrak{su}(N)$. The cubic Casimir operator $C_3 \equiv d_{abc}R^aR^bR^c$, commutes with all the $\mathfrak{su}(N)$ generators. Hence in light of Schur's lemma, $C_3$ is proportional to the $d \times d$ identity matrix. In particular, the cubic Casimir operator in the defining representation of $\mathfrak{su}(N)$ is given by

$$d_{abc}T^aT^bT^c = C_{3F}\mathbb{1}\,. \tag{28}$$

To evaluate $C_{3F}$, we multiply eq. (15) $d_{abd}$ to obtain

$$d_{abc}T^aT^b = \frac{N^2-4}{2N}T^c\,, \tag{29}$$

after using eqs. (26) and (27). Multiplying the above result by $T^c$ and employing eq. (4) yields

$$d_{abc}T^aT^bT^c = \frac{N^2-4}{2N}C_F\mathbb{1}\,. \tag{30}$$

Hence, using eqs. (5) and (28), we obtain

$$C_{3F} = \frac{(N^2-1)(N^2-4)}{4N^2}\,.$$

For completeness, we note the following result that resembles eq. (29),

$$f_{abc}T^aT^b = \tfrac{1}{2}\big(\{T^a,T^b\} + [T^a,T^b]\big) = \tfrac{1}{2}f_{abc}[T^a,T^b] = \tfrac{1}{2}if_{abc}f_{abd}T^d = \tfrac{1}{2}iNT^c\,,$$

after employing eq. (24). Hence, in light of eqs. (4) and (5) it follows that

$$f_{abc}T^aT^bT^c = \tfrac{1}{2}iNC_F\mathbb{1} = \tfrac{1}{4}i(N^2-1)\mathbb{1}\,.$$

Indeed, in any irreducible representation of $\mathfrak{su}(N)$, a similar analysis yields

$$f_{abc}R^aR^bR^c = \tfrac{1}{2}iNC_2\,, \tag{31}$$

where $C_2 \equiv R^aR^a$ is the quadratic Casimir operator in representation $R$. Hence, $f_{abc}R^aR^bR^c$ is proportional to $C_2$ and thus is not an independent Casimir operator.[4]

# 4 Matrices of the adjoint representation of SU($N$)

We now introduce the generators of $\mathfrak{su}(N)$ in the adjoint representation, which will be henceforth denoted by $F^a$. The $F^a$ are $(N^2-1) \times (N^2-1)$ antisymmetric matrices, since the dimension of the adjoint representation is equal to the number of generators of $\mathfrak{su}(N)$. Explicitly, the matrix elements of the adjoint representation generators are determined by the structure constants,

$$(F^a)_{bc} = -if_{abc}\,. \tag{32}$$

It is also convenient to define a set of $(N^2-1) \times (N^2-1)$ traceless symmetric matrices

$$(D^a)_{bc} = d_{abc}\,, \tag{33}$$

where the $d_{abc}$ is defined by eq. (17). Since $d_{abb} = 0$ it follows that $\mathrm{Tr}\,D^a = 0$. The properties of the $F^a$ and $D^a$ matrices have been examined in Refs. [3, 4].

---

[4]It may seem that eq. (30) implies that the cubic Casimir operator is proportional to the quadratic Casimir operator. However, the derivation of eq. (30) relies on eq. (15), which only applies to the generators of $\mathfrak{su}(N)$ in the defining representation. For an arbitrary $d$-dimensional irreducible representation of $\mathfrak{su}(N)$, $C_2$ and $C_3$ are generically independent.

The $F^a$ satisfy the commutation relations of the $\mathfrak{su}(N)$ generators,

$$[F^a, F^b] = i f_{abc} F^c\,, \tag{34}$$

which is equivalent to the Jacobi identity,

$$f_{abe} f_{ecd} + f_{cbe} f_{aed} + f_{dbe} f_{ace} = 0\,. \tag{35}$$

Likewise, there is a second commutation relation of interest,

$$[F^a, D^b] = [D^a, F^b] = i f_{abc} D^c\,, \tag{36}$$

which is equivalent to the two identities,

$$f_{abe} d_{cde} + f_{ace} d_{bde} + f_{ade} d_{bce} = 0\,, \tag{37}$$
$$f_{abe} d_{cde} + f_{cbe} d_{ade} + f_{dbe} d_{ace} = 0\,. \tag{38}$$

The relations,

$$F^a D^b + F^b D^a = D^a F^b + D^b F^a = d_{abc} F^c\,, \tag{39}$$

are also noteworthy. Combining eqs. (36) and (39) yields,

$$F^a D^b + D^a F^b = d_{abc} F^c + i f_{abc} D^c\,. \tag{40}$$

The expression for the commutator $[D^a, D^b]$ is more complicated,

$$\left[D^a, D^b\right]_{cd} = i f_{abe} (F^e)_{cd} - \frac{2}{N}\left(\delta_{ac}\delta_{bd} - \delta_{ad}\delta_{bc}\right)\,, \tag{41}$$

which is equivalent to the identity,

$$f_{abe} f_{cde} = \frac{2}{N}\left(\delta_{ac}\delta_{bd} - \delta_{ad}\delta_{bc}\right) + d_{ace} d_{bde} - d_{bce} d_{ade}\,. \tag{42}$$

Interchanging $b \leftrightarrow c$ and subtracting, the resulting expression can be rewritten as

$$(F^a F^b + D^a D^b)_{cd} = \frac{2}{N}\left(\delta_{ab}\delta_{cd} - \delta_{ac}\delta_{bd}\right) + d_{abe}(D^e)_{cd} + i f_{abe}(F^e)_{cd}\,. \tag{43}$$

Eq. (43) is equivalent to the identity,

$$f_{ace} f_{bde} - f_{abe} f_{cde} = \frac{2}{N}\left(\delta_{ab}\delta_{cd} - \delta_{ac}\delta_{bd}\right) + d_{abe} d_{cde} - d_{ace} d_{bde}\,. \tag{44}$$

The quadratic Casimir operator in the adjoint representation is

$$F^a F^a = C_A \mathbf{I}\,, \qquad \text{where } C_A = N\,, \tag{45}$$

and $\mathbf{I}$ is the $(N^2 - 1) \times (N^2 - 1)$ identity matrix, which is equivalent to eq. (26). Two other similar expressions of interest are

$$D^a D^a = \left(\frac{N^2 - 4}{N}\right)\mathbf{I}\,, \qquad F^a D^a = 0\,, \tag{46}$$

which are equivalent to eqs. (27) and (21), respectively.

Using the above results, we can derive additional identities of interest. For example,

$$f_{abc}F^bF^c = \tfrac{1}{2}f_{abc}\big[F^b,F^c\big] = \tfrac{1}{2}if_{abc}f_{bcd}F^d = \tfrac{1}{2}iNF^a, \tag{47}$$

$$f_{abc}F^bD^c = \tfrac{1}{2}f_{abc}\big[F^b,D^c\big] = \tfrac{1}{2}if_{abc}f_{bcd}D^d = \tfrac{1}{2}iND^a, \tag{48}$$

$$f_{abc}D^bD^c = \tfrac{1}{2}f_{abc}\big[D^b,D^c\big] = \tfrac{1}{2}i\left(f_{abc}f_{bcd} - \frac{4}{N}\delta_{ad}\right)F^d = i\left(\frac{N^2-4}{2N}\right)F^a. \tag{49}$$

It then follows that

$$f_{abc}F^aF^bF^c = \tfrac{1}{2}iN^2\mathbf{I}, \tag{50}$$

$$f_{abc}D^aF^bF^c = 0, \tag{51}$$

$$f_{abc}D^aD^bF^c = \tfrac{1}{2}i(N^2-4)\mathbf{I}, \tag{52}$$

$$f_{abc}D^aD^bD^c = 0. \tag{53}$$

For completeness, we quote the analogous identities with $f_{abc}$ replaced by $d_{abc}$. These identities are proved in Appendix B of these notes.

$$d_{abc}F^bF^c = \tfrac{1}{2}ND^a, \tag{54}$$

$$d_{abc}F^bD^c = \left(\frac{N^2-4}{2N}\right)F^a, \tag{55}$$

$$d_{abc}D^bD^c = \left(\frac{N^2-12}{2N}\right)D^a. \tag{56}$$

It then follows that

$$d_{abc}F^aF^bF^c = 0, \tag{57}$$

$$d_{abc}D^aF^bF^c = \tfrac{1}{2}(N^2-4)\mathbf{I}, \tag{58}$$

$$d_{abc}D^aD^bF^c = 0, \tag{59}$$

$$d_{abc}D^aD^bD^c = \left(\frac{(N^2-4)(N^2-12)}{2N^2}\right)\mathbf{I}. \tag{60}$$

Eq. (57) implies that the cubic Casimir operator in the adjoint representation vanishes.

Finally, we quote a number of useful trace identities [3–6].

$$\text{Tr}\,F^a = \text{Tr}\,D^a = 0, \qquad\qquad \text{Tr}(F^aD^b) = 0, \tag{61}$$

$$\text{Tr}(F^aF^b) = N\delta_{ab}, \qquad\qquad \text{Tr}(D^aD^b) = \left(\frac{N^2-4}{N}\right)\delta_{ab}, \tag{62}$$

$$\text{Tr}(F^aF^bF^c) = \tfrac{1}{2}iNf_{abc}, \qquad \text{Tr}(D^aF^bF^c) = \tfrac{1}{2}Nd_{abc}, \tag{63}$$

$$\text{Tr}(D^aD^bF^c) = i\left(\frac{N^2-4}{2N}\right)f_{abc}, \qquad \text{Tr}(D^aD^bD^c) = \left(\frac{N^2-12}{2N}\right)d_{abc}. \tag{64}$$

Additional identities involving traces of four generators can also be derived. Ref. [6] provides

the following results,[5]

$$\text{Tr}(F^a F^b F^c F^d) = \delta_{ad}\delta_{bc} + \tfrac{1}{2}(\delta_{ab}\delta_{cd} + \delta_{ac}\delta_{bd}) + \tfrac{1}{4}N(f_{ade}f_{bce} + d_{ade}d_{bce}), \tag{65}$$

$$\text{Tr}(F^a F^b F^c D^d) = \tfrac{1}{4}iN(d_{ade}f_{bce} - f_{ade}d_{bce}), \tag{66}$$

$$\text{Tr}(F^a F^b D^c D^d) = \tfrac{1}{2}(\delta_{ab}\delta_{cd} - \delta_{ac}\delta_{bd}) + \left(\frac{N^2-8}{4N}\right)f_{ade}f_{bce} + \tfrac{1}{4}N d_{ade}d_{bce}, \tag{67}$$

$$\text{Tr}(F^a D^b F^c D^d) = -\tfrac{1}{2}(\delta_{ab}\delta_{cd} - \delta_{ac}\delta_{bd}) + \tfrac{1}{4}N(f_{ade}f_{bce} + d_{ade}d_{bce}), \tag{68}$$

$$\text{Tr}(F^a D^b D^c D^d) = \frac{2i}{N}f_{ade}d_{bce} + i\left(\frac{N^2-8}{4N}\right)f_{abe}d_{cde} + \tfrac{1}{4}iN d_{abe}f_{cde}, \tag{69}$$

$$\text{Tr}(D^a D^b D^c D^d) = \left(\frac{N^2-4}{N^2}\right)\delta_{ad}\delta_{bc} + \left(\frac{N^2-8}{2N^2}\right)\delta_{ab}\delta_{cd} + \tfrac{1}{2}\delta_{ac}\delta_{bd} + \tfrac{1}{4}N f_{ade}f_{bce}$$
$$+ \left(\frac{N^2-16}{4N}\right)d_{ade}d_{bce} - \frac{4}{N}d_{abe}d_{cde}. \tag{70}$$

Alternative expressions for eqs. (65)–(70) are given in Appendix C [5].

As a check of eq. (65), let us set $a = c$ and sum over $a$. After employing eqs. (26) and (27) and relabeling $d$ by $c$, we obtain

$$\text{Tr}(F^a F^b F^a F^c) = \tfrac{1}{2}N^2\delta_{bc}. \tag{71}$$

Alternatively, one can obtain the above result directly by using eqs. (26), (45), (62) and (63) to compute

$$\text{Tr}(F^a F^b F^a F^c) = \text{Tr}\left((if_{abd}F^d + F^b F^a)F^a F^c\right) = if_{abd}\text{Tr}(F^d F^a F^c) + \text{Tr}(F^b F^a F^a F^c)$$
$$= if_{abd}\left(\tfrac{1}{2}iN f_{dac}\right) + N^2\delta_{bc} = \tfrac{1}{2}N^2\delta_{bc}, \tag{72}$$

which confirms the result of eq. (71). Similarly, the results of eqs. (66)–(70) can also be checked by multiplication by either a Kronecker delta, $f_{abc}$ or $d_{abc}$ and then employing the trace formulae previously derived.

Various applications of the identities given in this section can be found in a paper by Roger Cutler and Dennis Sivers [7]. Indeed, many of these identities are also reproduced in Appendix B of Ref. [7], after correcting the latter for some obvious typographical errors. The identities provided in these notes are sufficient to work out the color factors for scattering process involving quarks and gluons. Although the color factors should be computed for the case of $N = 3$, it is useful to first evaluate the color factors for an SU($N$) gauge theory, since these results allow one to identify sets of independent color factors that arise for a given process.

## 5 Two additional identities for $N = 3$

Two additional identities, which were first presented in Ref. [8], are special to the case of $N = 3$ and do not generalize to arbitrary $N$. These identities can be derived from the characteristic equation of a general element of the $\mathfrak{su}(3)$ Lie algebra [4,8],

$$\left\{F^a, F^b\right\}_{cd} = 3d_{abe}(D^e)_{cd} + \delta_{ab}\delta_{cd} - \delta_{ac}\delta_{bd} - \delta_{ad}\delta_{bc}, \tag{73}$$

$$\left\{D^a, D^b\right\}_{cd} = -d_{abe}(D^e)_{cd} + \tfrac{1}{3}\left(\delta_{ab}\delta_{cd} + \delta_{ac}\delta_{bd} + \delta_{ad}\delta_{bc}\right). \tag{74}$$

---

[5]In Ref. [6], the coefficient of $iN d_{abe}f_{cde}$ in eq. (69) is incorrectly given by $\tfrac{1}{2}$.

These two identities can be rewritten as

$$3d_{abe}d_{cde} - f_{ace}f_{bde} - f_{ade}f_{bce} = \delta_{ac}\delta_{bd} + \delta_{ad}\delta_{bc} - \delta_{ab}\delta_{cd}\,, \tag{75}$$

$$d_{abe}d_{cde} + d_{ace}d_{bde} + d_{ade}d_{bce} = \tfrac{1}{3}\big(\delta_{ab}\delta_{cd} + \delta_{ac}\delta_{bd} + \delta_{ad}\delta_{bc}\big)\,. \tag{76}$$

Combining eqs. (34) and (73) then yields,

$$(F^a F^b)_{cd} = \tfrac{1}{2}if_{abe}(F^e)_{cd} + \tfrac{3}{2}d_{abe}(D^e)_{cd} + \tfrac{1}{2}\big(\delta_{ab}\delta_{cd} - \delta_{ac}\delta_{bd} - \delta_{ad}\delta_{bc}\big)\,. \tag{77}$$

Likewise, combining eqs. (41) and (74) yields,

$$(D^a D^b)_{cd} = \tfrac{1}{2}if_{abe}(F^e)_{cd} - \tfrac{1}{2}d_{abe}(D^e)_{cd} + \tfrac{1}{6}\big(\delta_{ab}\delta_{cd} - \delta_{ac}\delta_{bd}\big) + \tfrac{1}{2}\delta_{ad}\delta_{bc}\,. \tag{78}$$

Note that the sum of eqs. (77) and (78) yields the $N = 3$ version of eq. (43). Unfortunately, there are no separate analogs of eqs. (77) and (78) for $N \neq 3$.

## A Traces of four generators in the defining representation of SU($N$)

The trace of a product of four generators in the defining representation also involves the symmetric tensor $d_{abc}$ introduced in Section 2. Applying eq. (15) twice, and taking the trace with the help of eq. (3) yields

$$\text{Tr}(T^a T^b T^c T^d) = \frac{1}{4N}\delta_{ab}\delta_{cd} + \tfrac{1}{8}\big(d_{abe}d_{cde} - f_{abe}f_{cde} + if_{abe}d_{cde} + if_{cde}d_{abe}\big)\,.$$

It is convenient to employ eqs. (38) and (42) of Section 3 to produce a more symmetric version,

$$\text{Tr}(T^a T^b T^c T^d) = \frac{1}{4N}\big(\delta_{ab}\delta_{cd} - \delta_{ac}\delta_{bd} + \delta_{ad}\delta_{bc}\big) + \tfrac{1}{8}\big(d_{abe}d_{cde} - d_{ace}d_{bde} + d_{ade}d_{bce}\big)$$
$$+ \tfrac{1}{8}i\big(d_{abe}f_{cde} + d_{ace}f_{bde} + d_{ade}f_{bce}\big)\,. \tag{79}$$

A nice check of eq. (79) is to rederive eq. (14) by setting $a = c$ and summing over $a$.

## B Proof of eqs. (54)–(56)

First, we note that eqs. (54)–(56) are equivalent to the last three trace identifies of eqs. (63) and (64),

$$\text{Tr}(D^a F^b F^c) = d_{ade}(F^d F^e)_{bc}\,, \tag{80}$$

$$\text{Tr}(D^a D^b F^c) = d_{ade}(F^d D^e)_{bc}\,, \tag{81}$$

$$\text{Tr}(D^a D^b D^c) = d_{ade}(D^d D^e)_{bc}\,, \tag{82}$$

after using eqs. (32) and (33). Multiplying eq. (40) on the left by $F^e$ and taking a trace yields

$$\text{Tr}(F^e F^a D^b) = \tfrac{1}{2}N d_{abe}\,, \tag{83}$$

in light of eqs. (61) and (62). Likewise, multiplying eq. (40) on the right by $D^e$ and taking a trace yields

$$\text{Tr}(F^a D^b D^e) = \frac{i(N^2 - 4)}{2N}f_{abe}\,. \tag{84}$$

Multiplying eq. (43) on the right by $(D^f)_{de}$ and taking the trace (by setting $c = e$ and summing over $e$) yields,

$$\text{Tr}(F^a F^b D^f + D^a D^b D^f) = \left(\frac{N^2 - 6}{N}\right) d_{abf}. \tag{85}$$

Finally, we use the result of eq. (83) to obtain

$$\text{Tr}(D^a D^b D^f) = \left(\frac{N^2 - 12}{2N}\right) d_{abf}. \tag{86}$$

## C   Traces of adjoint representation matrices revisited

The traces of products of four matrices (either $F^a$ or $D^a$) in the adjoint representation are given in eqs. (65)–(70). It is sometime convenient to eliminate the product $f_{ade}f_{bce}$ in favor of $\delta_{ab}$ and $d_{abc}$, etc., by using eq. (42). The following results were obtained in Ref. [5],

$$\text{Tr}(F^a F^b F^c F^d) = \delta_{ab}\delta_{cd} + \delta_{ad}\delta_{bc} + \tfrac{1}{4}N\big(d_{abe}d_{cde} - d_{ace}d_{bde} + d_{ade}d_{bce}\big),$$

$$\text{Tr}(F^a F^b F^c D^d) = \tfrac{1}{4}iN(d_{abe}f_{cde} + f_{abe}d_{cde}),$$

$$\text{Tr}(F^a F^b D^c D^d) = \left(\frac{N^2 - 4}{N^2}\right)\big(\delta_{ab}\delta_{cd} - \delta_{ac}\delta_{bd}\big) + \left(\frac{N^2 - 8}{4N}\right)\big(d_{abe}d_{cde} - d_{ace}d_{bde}\big) + \tfrac{1}{4}N d_{ade}d_{bce},$$

$$\text{Tr}(F^a D^b F^c D^d) = \tfrac{1}{4}N\big(d_{abe}d_{cde} - d_{ace}d_{bde} + d_{ade}d_{bce}\big),$$

$$\text{Tr}(F^a D^b D^c D^d) = i\left(\frac{N^2 - 12}{4N}\right)f_{abe}d_{cde} + \frac{i}{N}\big(f_{ade}d_{bce} - f_{ace}d_{bde}\big) + \tfrac{1}{4}iN d_{abe}f_{cde},$$

$$\text{Tr}(D^a D^b D^c D^d) = \left(\frac{N^2 - 4}{N^2}\right)\big(\delta_{ab}\delta_{cd} + \delta_{ad}\delta_{bc}\big) + \left(\frac{N^2 - 16}{4N}\right)\big(d_{abe}d_{cde} + d_{ade}d_{bce}\big) - \tfrac{1}{4}N d_{ace}d_{bde}.$$

Note that the second equation above is consistent with eq. (66) in light of eq. (36), and the fifth equation above is consistent with eq. (69) in light of eq. (37).

## Acknowledgments

I am grateful to Peter Arnold and Eugene Kogan, who encouraged me to make these notes more widely available. This work is supported in part by the U.S. Department of Energy grant number DE-SC0010107.

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
