# Peer review of "Useful relations among the generators in the defining and adjoint representations of SU(N)"

_SciPost Physics Lecture Notes, doi:SciPost Phys. Lect. Notes 21 (2021)_

## Round 1 · Referee Report · Anonymous (Referee 1) · 2020-12-28

Strengths
1-The writer is clear in their goal and presentation of their work
2-Many identities that require long derivation are presented in an easy to understand and compact manner
3-Identities presented in this work are easily applied to problems in physics. In particularly, trace identities which appear when working with chiral Lagrangians in quantum field theory come to mind.
2-Many identities that require long derivation are presented in an easy to understand and compact manner
3-Identities presented in this work are easily applied to problems in physics. In particularly, trace identities which appear when working with chiral Lagrangians in quantum field theory come to mind.
Weaknesses
1-There do not appear to be any new results in this work. The point of this work is to collect results in the literature and present them in a succinct way.
Report
Assuming that this work is correct in its assertion that there is no single reference compiling the identities, I find that this work meets the criteria for publishing in SciPost Physics Lecture Notes. The work is clear and a useful resource for anyone working with the generators of SU(N). I was able to reproduce all equations just by following the work.
Requested changes
I have no requested changes.

---

## Editorial Decision

published